# Evolutionary Steps in the Emergence of Life Deduced from the Bottom-Up Approach and GADV Hypothesis (Top-Down Approach)

**DOI:** 10.3390/life6010006

**Published:** 2016-01-26

**Authors:** Kenji Ikehara

**Affiliations:** 1G & L Kyosei Institute, Keihannna Labo-401, Hikaridai 1-7, Seika-cho, Sorakugun, Kyoto 619-0237, Japan; ikehara@cc.nara-wu.ac.jp; Tel./Fax: +81-774-73-4478; 2International Institute for Advanced Studies of Japan, Kizugawadai 9-3, Kizugawa, Kyoto 619-0225, Japan

**Keywords:** origin of life, [GADV]-protein world, GADV hypothesis, pseudo-replication, protein 0^th^-order structure

## Abstract

It is no doubt quite difficult to solve the riddle of the origin of life. So, firstly, I would like to point out the kinds of obstacles there are in solving this riddle and how we should tackle these difficult problems, reviewing the studies that have been conducted so far. After that, I will propose that the consecutive evolutionary steps in a timeline can be rationally deduced by using a common event as a juncture, which is obtained by two counter-directional approaches: one is the bottom-up approach through which many researchers have studied the origin of life, and the other is the top-down approach, through which I established the [GADV]-protein world hypothesis or GADV hypothesis on the origin of life starting from a study on the formation of entirely new genes in extant microorganisms. Last, I will describe the probable evolutionary process from the formation of Earth to the emergence of life, which was deduced by using a common event—the establishment of the first genetic code encoding [GADV]-amino acids—as a juncture for the results obtained from the two approaches.

## 1. Introduction

In order to solve the riddle on the origin of life, it is important to understand the process by which the fundamental life system was formed, which is composed of three major elements: genes, genetic code, and proteins. However, it seems that even researchers studying the origin of life have not paid much attention to the formation process of the system, probably because it is quite difficult to even start to understand the process and because, when researchers address this problem, they would generally first try to understand what happened on the primitive Earth, where the first life emerged. Actually, many studies on the origin of life have been carried out from such a standpoint, the bottom-up approach, which has been intended to make clear the steps extending from the formation of Earth to the emergence of life (Figure 1). The bottom-up approach is surely a general way for studying the origin of life, but it is incapable of solving this problem alone, principally because it is not able to elucidate the formation process of the fundamental life system composed of genes, genetic code, and proteins.

On the other hand, the RNA world hypothesis has been the main focus for studies on the origin of life, since the RNA world hypothesis was proposed by W. Gilbert [1] to solve the so-called “chicken-egg relationship” between genes and proteins just after several catalytic RNAs or ribozymes were discovered [2,3]. The hypothesis assumes that the RNA world was first formed by self-replication of catalytic RNA(s) and, later, genetic information and catalytic function on the RNA were transferred to DNA and protein, respectively.

**Figure 1 life-06-00006-f001:**
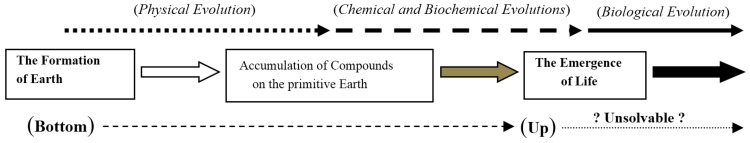
General approach for studies on the origin of life (bottom-up approach), which are carried out to elucidate what happened on the primitive Earth, where the first life emerged. Molecular evolution from simple inorganic compounds would occur in order of physical evolution (dotted arrow), chemical and biochemical evolutions (broken arrow), and biological evolution (solid arrow). The difficulty in understanding the evolutionary processes with the bottom-up approach is shown through the order of the thick arrows, from white to gray to black. The thin broken arrow represents the direction of the study with bottom-up approach.

However, there are many weak points in this hypothesis. For example, (1) nucleotides have not been produced from simple inorganic compounds through prebiotic means and have not been detected in any meteorites, although a small quantity of nucleobases can be obtained [4]. (2) It is quite difficult or most likely impossible to synthesize nucleotides and RNA through prebiotic means. (3) It must also be impossible to self-replicate RNA with catalytic activity on the same RNA molecule. (4) It would be impossible to explain the formation process of genetic information according to the RNA world hypothesis, because the information is comprised of triplet codon sequence, which would never be stochastically produced by joining of mononucleotides one by one. (5) The formation process of the first genetic code cannot be explained by the hypothesis either, because a genetic code composed of around 60 codons must be prepared to synthesize proteins from the beginning. (6) It is also impossible to transfer catalytic activity from a folded RNA ribozyme to a protein with a tertiary structure.

Thus, it is only a matter of time before it is recognized that it must be impossible to solve the riddle of the origin of life from the standpoint of the RNA world hypothesis. Actually, some researchers have started to search for new ways to solve the riddle of the origin of life, especially in recent years. For example, co-evolution theory between peptides and RNA has been provided by B.R. Francis, in order to reinforce the instability of RNA by hypothesizing that the earliest predecessor of the nucleic acids was a β-linked polyester made from malic acid [5]. Restoration from the gene-early theory into the metabolism-early theory has also been considered by P.T.S. van der Gulik and D. Speijer [6,7]. Caetano-Anollés’s group suggests that primordial metabolic domains evolved earlier than informational domains involved in translation and transcription, supporting the metabolism-first hypothesis rather than the RNA world scenario [8].

I consent to their ideas because it is important to suppose a stable predecessor of RNA or to stabilize fragile RNA with peptides, especially in the watery circumstances on the primitive Earth, and RNA having a complex chemical structure would not be synthesized without a metabolism using catalytic peptides and/or proteins as their aggregates. However, it must also be impossible to solve the riddle of the origin of life even using both the co-evolution theory and the metabolism-early theory, since genetic information composed of triplet base sequences would never be stochastically formed on the RNA, which is synthesized by joining mononucleotides one by one—even if peptides could stabilize the RNA sequence and a metabolism with peptide catalysts could assist to accumulate RNA on the primitive Earth. In fact, Kurland is also considering that RNA coding is not a *sine qua non* for the accumulation of catalytic polypeptides [9]. It is also proposed by Maury that the amyloid world—which would be stable, self-replicative, environmentally responsive, and evolvable under early Earth conditions—preceded the RNA world, since it is unlikely that functional RNA could have existed under the presumably very harsh conditions on the early Earth [10,11]. In addition, Carter and Wächtershäuser also give criticisms against the RNA world theory [12,13].

Under such a recent situation, a question, “Why is the origin of life still a mystery?”, was given in a workshop of OQOL 2014, which was held at the International Institute for Advanced Studies of Japan, Kyoto, Japan, 12–13 July 2014. In the workshop, P.L. Luisi [14] presented a lecture entitled “The need of a new start from ground zero,” and S. Benner [15] provided five paradoxes on the origin of life. Therefore, it is obvious that the riddle of the origin of life still remains unanswered in spite of the strenuous efforts made by many researchers working in the field. Thus, I believe that elucidation of the riddle will be delayed if researchers continue to persist with the RNA world hypothesis.

Then, where do we start in solving the riddle of the origin of life? Adoption of a new approach that nobody has tried so far could lead to solving the riddle of the origin of life. This new approach might be a top-down approach, studying first the fundamental genetic system of extant organisms and the subsequent emergence of life (Figure 2).

**Figure 2 life-06-00006-f002:**
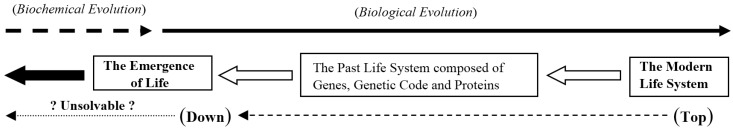
Top-down approach carried out to highlight the direction of evolution of the modern life system into the emergence of life. Molecular evolution would occur in order of biochemical evolution (broken arrow) and biological evolution (solid arrow). It becomes difficult to make clear the steps with the top-down approach in the order of the biological and biochemical evolutions, shown by white and black thick arrows, respectively. The thin broken arrow represents the direction of the study with the top-down approach.

In this review, I would like to propose that cooperation of two counter-directional approaches would be effective in solving the riddle on the origin of life, and I will describe probable steps in the emergence of life deduced from the outline in the last section of this review.

## 2. General Approach for the Study of the Origin of Life (Bottom-Up Approach)

### 2.1. Chemical Analysis of Extracts from Old Rocks and Meteorites

Many researchers analyzed chemically organic compounds in extracts from rocks and fossils on the primitive Earth and investigated extracts of carbonaceous meteorites from space, which might provide insight into the situation of the primitive Earth. Thus, many early works were carried out from the standpoint of the bottom-up approach, and important experimental results have been obtained, as follows:
Twelve amino acids were found from about 3.1 billion-year-old pre-Cambrian Fig Tree Chart from the mining area of eastern Transvaal, South Africa [16]. Extracts from 3.8 billion-year-old Isua rock formed on the primitive Earth were also chemically analyzed, and both simple amino acids and hydrocarbons were detected from the extracts [17]. However, Nagy *et al.* have concluded, based on the extent of amino acid racemization, that the amino acids may be modern to a few tens of thousands of years old [17]. Van Zuilen *et al.* also asserted that previously presented evidence for ancient traces of life in the highly metamorphosed Early Archaean rock, or the metasomatic rock records, should be reassessed, which were earlier thought to provide the basis for inferences about early life [18].Several amino acids were detected in extracts of meteorites from space, such as Marchison meteorite, which may be similar to rocks on the primitive Earth [19]. Therefore, nowadays, amino acids identified in the Murchison chondroritic meteorite are thought to have been delivered to the early Earth by meteorites, asteroids, comets, and interplanetary dust particles, which may trigger the appearance of life by assisting in the synthesis of proteins via prebiotic polymerization reactions [20,21,22].


### 2.2. Physical Evolution Experiments

Molecular evolution from simple inorganic compounds would occur in order of physical evolution, chemical evolution, and biochemical evolution to produce biologically important organic compounds, such as amino acids and peptides. However, inorganic compounds in the primitive atmosphere cannot generally react to produce organic compounds if energy is not supplied, because of their poor reactivities. Therefore, physical evolution experiments were carried out using various energy suppliers.
About 60 years ago, Miller published results showing that amino acids and nucleobases were synthesized by repeated electrical discharging into a reducing gas mixture containing CH_4_, NH_3_, H_2_, and H_2_O, which imitates lightning in the primitive atmosphere [4,23,24,25]. Many geologists today consider that the early atmosphere was rather weakly reducing or even neutral, composed of mainly CO_2_ and N_2_, based on later studies on the primitive atmosphere [26]. However, it has been confirmed that significant amounts of amino acids can still be synthesized even with weakly reducing or neutral primitive atmospheric gas [27,28].It is supposed that deep-sea hydrothermal environments were important sites for the synthesis of bioorganic molecules, leading to the emergence of life [29]. Simple amino acids were synthesized with experimental equipment mimicking hydrothermal vents in a deep sea on the primitive Earth [30]. It has been also demonstrated that oligopeptides were synthesized from glycine in a flow reactor simulating a submarine hydrothermal system [31,32,33].Oró *et al.* suggested that cometary collisions with the primitive Earth provided the planet with both free energy and volatiles as important sources for creation of transient, gaseous environments, in which prebiotic synthesis may have taken place [21]. Experiments reproducing meteorite impacts or heavy bombardments to the primitive Earth were also carried out. After the impact, numerous organic molecules, including fatty acids, amines, nucleobases, and amino acids were detected. So, it is considered that organic molecules on the early Earth may have arisen from such impact syntheses [34,35].

### 2.3. Planetary Exploration and Astronomical Observation

The cosmic ancestry hypothesis—like the Panspermia hypothesis, assuming that life on Earth was seeded from space—has been proposed as an explanation of the origin and evolution of life on Earth. In recent years, the idea has been expanded to include the introduction of organic compounds as amino acids from space onto the primitive Earth. Such being the case, nowadays, many investigations—for example, space exploration of planets, asteroids, satellites, and comets with a probe and observation of them using various kinds of radio telescopes—have been prosperously carried out as one main current of studies for solving the riddle of the origin of life.

Actually, sand and dust, which were carried back by space exploration of the asteroid Itokawa, were chemically analyzed [36]. Scientific analyses have been also carried out by instruments on the probe, which were soft-landed onto the comet Churyumov–Gerasimenko [37], because the comet may have some analogy with the primitive Earth. Ethylene oxide and acetaldehyde were detected through multiple lines in the hot cores NGC 6334F, G327.3 − 0.6, G31.41 + 0.31, and G34.3 + 0.2, as shown in the results of spectrum analysis using a telescope of the atmosphere of planets similar to the primitive Earth [38].

However, any convincing facts that explain the origin of life have not been discovered by the space extrapolations and astronomical observations so far.

### 2.4. Chemical Evolution Experiments

A variety of organic compounds including [GADV]-amino acids, Gly [G], Ala [A], Asp [D], and Val [V] could be synthesized by lightning in the primitive atmosphere and by comet and/or meteorite impacts on the surface of the primitive Earth. In addition, those compounds were also delivered by meteorites and asteroids, as described above. So, it is considered that amino acids were naturally abundant in various prebiotic contexts [39].

Wächtershäuser also reported that many kinds of organic compounds including aspartic acid would be produced by pyrite-pulled reduction in the presence of H_2_S and NH_3_ and accumulated on the primitive Earth [40]. Peptides could be produced with amino acids accumulated on the planet in the hot and salty environments of the primordial earth, as shown with the salt-induced peptide formation in a hot prebiotic ocean in the presence of metal ions as Cu^2+^ [41,42,43].

### 2.5. Biochemical Evolution Experiments

It has been confirmed that some peptides have various kinds of catalytic activities. For example, not only dipeptide, Ser-His, but also Gly-Gly and Gly-Gly-Gly catalyze peptide bond formation between amino acids, although less efficiently [44]. It has also been reported that simple peptides, especially His-containing peptides, could polymerize not only amino acids but also nucleotides into short oligomers under plausible prebiotic conditions [41,45].

However, Ser-His dipeptide could not play any role in biochemical evolutionary process on the primitive Earth, even if the dipeptide has a high catalytic activity, because it would be hard to obtain the dipeptides on the primitive Earth due to the low abundance of His. On the contrary, Gly-Gly dipeptide and Gly-Gly-Gly tripeptide would play a role, even if these peptides have only a low catalytic activity. The low activity of the simple peptide catalysts is out of the question, because it is sufficient for synthesis of organic compounds in the absence of any effective proteineous catalyst on the primitive Earth.

#### 2.5.1. Catalytic Activities of [GADV]-Peptides Produced by Repeated Heat-Drying Processes

[GADV]-P30 was produced by repeated heat-drying [GADV]-amino acid solution 30 times [42]. The repeated heat-drying cycles were carried out to mimic [GADV]-peptide synthesis in depressions on rocks or tide pools on the primitive Earth. As the number of the cycles increases, yellowish fluorescent light became strong, suggesting that some cyclic compounds were formed [46].

[GADV]-P30 hydrolyzed β-galactoside bond in MetU-Gal and amide bond (peptide bond) in Gly-pNA, resulting in production of MetU and pNA, respectively. The [GADV]-P30 did hydrolyze a peptide bond in a natural protein, bovine serum albumin (BSA) [46]. All three catalytic activities examined with [GADV]-P30 were detected, suggesting that [GADV]-peptides and/or [GADV]-proteins as their aggregates could exhibit various catalytic activities.

#### 2.5.2. Catalytic Activities of [GADV]-Random Octapeptides

[GADV]-octapeptides, which were randomly joined with solid-phase peptide synthesis, hydrolyzed BSA to give several fragments of the protein. In addition, the hydrolytic activity was much higher than that of [GADV]-P30 prepared by the repeated heat-drying treatment [46]. This indicates that even [GADV]-peptides without cyclic compounds such as diketopiperazine, which is formed during the repeated dry-heating process, can exhibit catalytic function of hydrolysis toward peptide bonds. The higher hydrolytic activity of the random octapeptides suggests that the appearance of [GADV]-peptides/proteins produced after the GNC primeval genetic code could cause the steps to the emergence of life to proceed at a higher rate than before establishment of the primeval genetic code.

### 2.6. Limitations of Bottom-Up Approaches

The bottom-up approaches, which are generally carried out by experiments, are undoubtedly important to understand how life emerged on the primitive Earth. Indeed, the important insight that [GADV]-amino acids would accumulate on the primitive Earth was obtained through experiments [47,48]. The evolutionary processes deduced from the results obtained by the bottom-up approaches are shown in Figure 3. However, on the other hand, it would be quite difficult or most likely impossible to solve the riddle of the origin of life using only bottom-up approaches. The principal reason is because it is impossible to know how the fundamental life system, which is composed of genes, genetic code, and proteins, was formed. Of course, this would not be the case if vestiges of newly-born lives possessing a primordial life system were discovered from an old rock, meteorite, or hydrothermal vent, and so on. However, no vestige has been discovered so far. It might also be problematic that the bottom-up approach tends to focus on experiments and, inevitably, lacks theoretical considerations.

**Figure 3 life-06-00006-f003:**
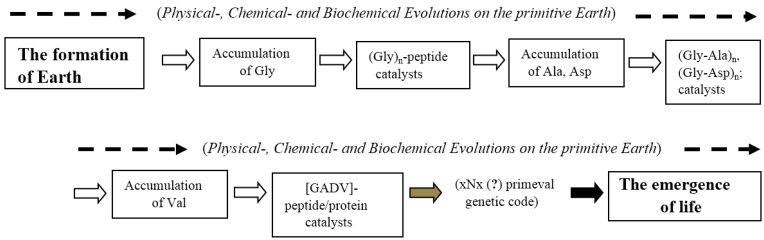
Summarized results obtained by the bottom-up approach for study on the origin of life. Physical, chemical, and biochemical evolutions indicated by broken arrows would take place in parallel in some cases from the accumulation of glycine (Gly) to the emergence of life. The steps made clear with the bottom-up approach are shown by thick white arrows. Assumed and unknown steps are shown with thick gray and black arrows, respectively.

## 3. A Top-down Approach for Understanding the Origin of Life ([GADV]-Protein World Hypothesis)

About 20 years ago, I thought of studying the mechanism behind how entirely new genes have been formed in presently existing microorganisms. That, fortunately, led me to research the origin of life. Consequently, I adopted the top-down approach for the research by chance. The progress of my study is described briefly below.

### 3.1. Origin of Entirely New Genes

I started a study to understand how entirely new genes—*i.e*., the first ancestor genes in families—are formed on the present Earth, independent of the origin of life. For that purpose, data on microbial water-soluble globular proteins and their genes obtained from GenomeNet Databases were analyzed using six property conditions (hydropathy, α-helix, β-sheet and turn/coil structure formations, acidic amino acid, and basic amino acid compositions). The four protein structural indexes for hydrophobicity, α-helix, β-sheet, and turn/coil formations were calculated by summation using the following Equation; *I*(*x*)*_t_* = Σ*I*(*x*)*_a_ n_a_/n_t_*, where *I*(*x*)*_t_*, *I*(*x*)*_a_*, *n_a_*, and *n_t_* are total index of a protein, index for each amino acid, each amino acid number and total amino acid number in a protein, respectively [49]. Necessary hydropathy index and secondary structure indexes of an amino acid were obtained from Stryer’s textbook [50]. Acidic (aspartic and glutamic acids) and basic (histidine, lysine, and arginine) amino acid contents were arithmetically calculated. From the results, it was found that the six structural indexes of water-soluble globular protein obtained with data of seven microbial genomes (*Mycobacterium tuberculosis*, *Aeropyrum pernix*, *Escherichia coli*, *Bacillus subtilis*, *Haemophilus influenzae*, *Methanococcus genitarium*, and *Borrelia burgdorferi*) having a different GC content from each other were all roughly constant, irrespective of a significant change of the GC content of a gene [49,51]. So, the six structural indexes were used as the conditions for judging whether a polypeptide can fold into a water-soluble globular structure.

Furthermore, it was found that an imaginary protein encoded by a non-stop frame on antisense strand of GC-rich gene (GC-NSF(a)) satisfies the six conditions at a high probability. It was also confirmed that non-stop frame is produced at a high probability, since three termination codons are AT-rich and GC-NSF(a) is literally GC-rich [49,51,52]. This means that all imaginary proteins, which are encoded by any GC-NSF(a), fold into a water-soluble globular protein structure, although they have different amino acid sequences, respectively. One of the reasons is because rather similar or symmetrical base compositions at three codon positions are observed between GC-rich gene on a sense strand and antisense sequence of GC-rich gene (GC-NSF(a)) [49,52,53]. Thus, I finally reached the conclusion that entirely new genes are produced from non-stop frames on antisense strands of, not AT-rich, but GC-rich microbial genes [49,52,53,54,55].

### 3.2. Origin of the Genetic Code

In the process of analyzing base compositions at three codon positions of GC-rich genes and the corresponding antisense sequences, I noticed that the base composition formats of both highly GC-rich genes (65%–70%) and GC-NSF(a) sequences are roughly SNS, where S means guanine: G, or cytosine: C. Then, it was confirmed by computer analysis that imaginary proteins produced under SNS code encoding 10 amino acids ([GADV]-amino acids plus Glu [E], Leu [L], Pro [P], His [H], Gln [Q], and Arg [R]) satisfy the six conditions for formation of a water-soluble globular structure. At this time, we published the SNS primitive genetic code hypothesis [56].

Further, we looked for a minimum set of amino acids that could produce proteins satisfying four conditions (hydropacy plus three formabilities of α-helix, β-sheet, and turn/coil), in order to confirm whether there existed a genetic code more ancient than the SNS code, or not. From the results, it was found that [GADV]-proteins encoded by GNC code satisfy the four conditions, when about equal amounts of [GADV]-amino acids are contained in the proteins [49]. This means that a group of four [GADV]-amino acids could produce proteins basically comparable to contemporary proteins.

Although [GADV]-proteins do not contain basic amino acid, divalent metal ions, such as Mg^2+^, Mn^2+^ and Cu^2+^, could compensate for the lack of positive charge in the acidic [GADV]-protein. In this way, I reached a GNC-SNS primitive genetic code hypothesis, suggesting that the universal or standard genetic code originated from GNC code encoding four [GADV]-amino acids, through SNS code (Figure 4) [49].

**Figure 4 life-06-00006-f004:**
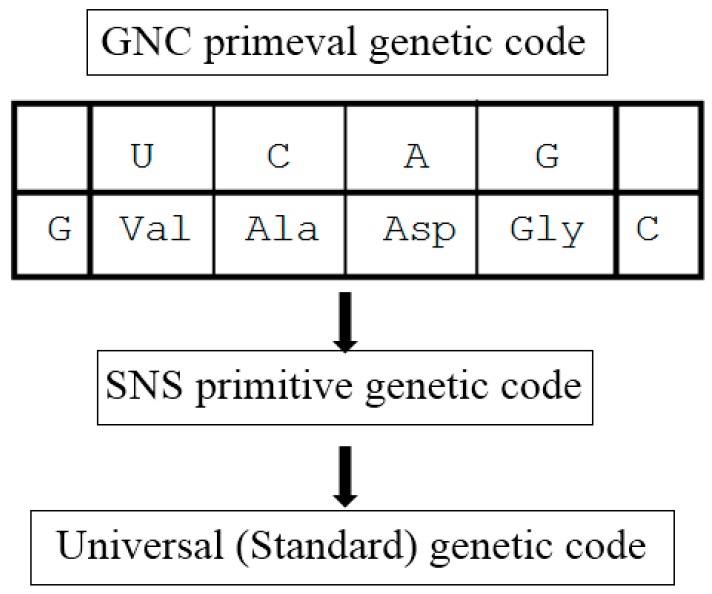
GNC-SNS primitive genetic code hypothesis, assuming that the universal or standard genetic code (both formally and substantially triplet code) originated from GNC primeval genetic code (formally triplet but substantially singlet code) through SNS primitive genetic code composed of 10 amino acids encoded by 16 codons (formally triplet but substantially doublet code).

This indicates that the frameworks or protein 0^th^-order structures for creation of entirely new proteins are implicitly written in the genetic code table. The GNC-SNS hypothesis is quite important for the elucidation of the origin of life too, since elucidation of the origin of the genetic code, which is located at a core position connecting genetic information on a nucleotide sequence of RNA and/or DNA with the catalytic function of protein, could lead to making clear the origins of genes and proteins.

### 3.3. [GADV]-Protein World Hypothesis: GADV Hypothesis

I had considered about 15 years ago that the first proteins must be composed of four simple [GADV]-amino acids, if the GNC-SNS primitive genetic code hypothesis is correct. At that time, I suddenly came across the [GADV]-protein world hypothesis on the origin of life or the GADV hypothesis. In other words, I noticed that [GADV]-proteins could be pseudo-replicated through [GADV]-protein synthesis with a [GADV]-protein having catalytic activity for peptide bond formation, suggesting that various water-soluble globular [GADV]-proteins could be produced by the pseudo-replication. The term “pseudo-replication” means a process whereby proteins comprised of the same constituent set of amino acids (composition) that possess similar but different water-soluble globular structures are generated by a random process without resorting to any genetic system [57]. The fact that even simple Gly-Gly dipeptides can catalyze peptide bond formation supports the idea of this process [44]. The notion of the random polymerization of [GADV]-amino acids without any genetic system led to a new theory on the origin of life: the GADV hypothesis, assuming that life originated from a [GADV]-protein world, not from the RNA world as previously thought [51,57,58]. Thus, I formed an original idea, the GADV hypothesis, which was obtained through studies on the formation process of the fundamental life system from the present to the past, going back in a time lapse or with a top-down approach (Figure 5) [57,58,59].

**Figure 5 life-06-00006-f005:**
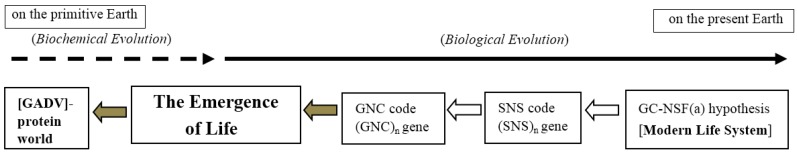
Summarized results obtained by top-down approach for study of the origin of life. Periods of biochemical and biological evolution are shown by broken and solid arrows, respectively. The steps made clear with the top-down approach are shown with thick white arrows. Steps deduced from the top-down approach are shown with thick gray arrows in the figure.

The probable evolutionary process from a [GADV]-protein world through the emergence of life to the modern life system, deduced from a GADV hypothesis, are shown in Figure 6. The evolutionary steps from accumulation of nucleotides to formation of ds-(GNC)_n_ genes are only assumed without experimental results at this time. However, the steps follow a logical sequence, if the GNC primeval genetic code hypothesis is correct. The first reason is because GNC code must be established before formation of (GNC)_n_ genes, since any gene cannot be expressed without genetic code. The second reason is because the genetic information composed of triplet codon sequence cannot be stochastically produced by joining of mononucleotides one by one. The latter reason suggests that the first genetic information must be created by joining GNC sequences. I would like to stress here that my propositions are based on analyses of databases of genes and proteins. So, I believe that there is no way to explain the emergence of life, except for the steps deduced from the GADV hypothesis.

**Figure 6 life-06-00006-f006:**
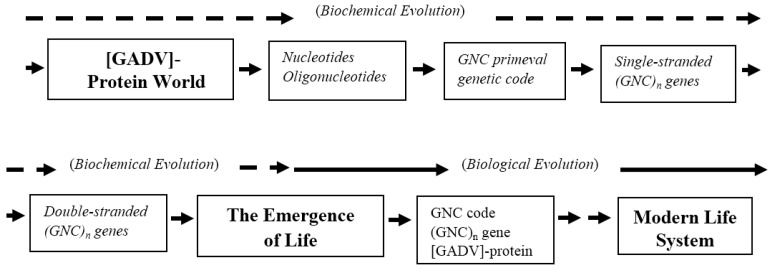
The evolutionary process from formation of a [GADV]-protein world to the modern life system through the emergence of life, which is deduced from the GADV hypothesis. Periods of biochemical and biological evolutions are indicated by broken and solid arrows. Italic letters in boxes represent evolutionary steps assumed from the GADV hypothesis.

### 3.4. Strengths of the GADV Hypothesis

I have introduced two new concepts for proposing the GADV hypothesis: protein 0^th^-order structure and pseudo-replication of [GADV]-protein [51,57,58,59,60,61]. Owing to the introduction of the two concepts, a novel idea about the origin of life, the GADV hypothesis, emerged. Based on the GADV hypothesis, the process by which the fundamental life system composed of genes, genetic code, and proteins was formed can be understood without any major contradictions. In addition, the formation process of the “chicken and egg” relationship between gene and protein can be also explained according to the GADV hypothesis, as moving from the lower ([GADV]-protein synthesis) to the upper stream (creation of (GNC)_n_ genes) of the genetic flow in the fundamental life system [57,59].

### 3.5. Limitations of the GADV Hypothesis

Although there are many strong points in the GADV hypothesis as described above, there are several weak points in the hypothesis, too. The biggest weakness is that only limited experimental results have been obtained thus far, such as detection of enzymatic activities of [GADV]-peptides/proteins as aggregates and of random [GADV]-octapeptides, for hydrolysis of peptide bond in a natural protein, bovine serum albumin (BSA) [46]. The reason is partly because we have mainly concentrated on the studies using computational analysis of databases of microbial genes and proteins. However, I would like to emphasize here that the hypothesis is not a purely theoretical idea, since the hypothesis was based on databases, which were obtained by experiments, and is based on protein 0^th^-order structure as [GADV]-amino acids, which satisfies the four structural conditions obtained from analysis of experimental data of proteins in extant micro-organisms [51,58,61].

However, it is difficult to get a clear answer for the origin of life according to only the GADV hypothesis or the top-down approach, because we are not able to fundamentally understand how, and from what kinds of simple inorganic and organic compounds, [GADV]-amino acids and [GADV]-proteins were synthesized and accumulated on the primitive Earth (Figure 2).

## 4. The Necessity of Combining the Bottom-Up and Top-Down Approaches

It is obviously quite difficult to solve the riddle of the origin of life only from the top-down approach or only from the bottom-up approach. However, all events from the formation of Earth through to the emergence of life and the evolution of modern lives occurred on a straight timeline. Therefore, in order to overcome the difficulties in elucidating the origin of life, it is essential to take the results obtained from the two counter-directional approaches into consideration and to search for consistent evolutionary processes leading to the present life system from accumulation of simple organic compounds on the primitive Earth. Then, it is also important to understand the establishment process of the first genetic code, GNC, because tRNA realizing genetic code connects the genetic function expressed by anticodon of tRNA with the catalytic function of proteins synthesized using amino acids on the tRNA.

### 4.1. The Most Significant Evolutionary Steps according to Bottom-Up Approaches

As described in Section 2, it was confirmed that endogenous, exogenous, and impact-shock sources of organics like amino acids could each have made a significant contribution to the origins of life. Although, of course, which organic compounds were synthesized with prebiotic means at quantitatively high amounts depends on primitive Earth’s conditions, such as the composition of the early terrestrial atmosphere [22]. Van der Gulik *et al.* have indicated that the [GADV]-amino acids are abundantly produced in Miller-type prebiotic synthesis experiments, and that [GADV]-amino acids are contained abundantly in carbonaceous meteorites [47]. Therefore, they have conjectured that the early functional peptides were made of [GADV]-amino acids. Parker *et al.* reported that the overall abundances of the synthesized amino acids in the presence of H_2_S are very similar to those found in some carbonaceous meteorites [28]. Higgs concluded that [GADV]-amino acids were used to synthesize early peptide catalysts [48], and proposed a four-column hypothesis, xNx code encoding four [GADV]-amino acids. The small letter x and the capital letter N mean all and any one of four nucleobases (A, U, G, and C), respectively. Francis suggests that the singlet coding system evolved into a four nucleotide/four amino acid process (AMP = aspartic acid, GMP = glycine, UMP = valine, CMP = alanine) [5] and Di Giulio insists that the universal genetic code originated from a primitive code, GNS and GNN, encoding four [GADV]-amino acids [62]. Certainly, there are some slight differences among those ideas. However, it is important to note that the common conclusion—that the first proteins were produced with [GADV]-amino acids—has been independently obtained with bottom-up approaches.

Van der Gulik *et al.* [47] assumed that some traces of prebiotic peptides composed of [GADV]-amino acids still exist in the form of active sites in present-day proteins. After searching for such proteins as prebiotic peptide candidates, they identified three main classes of motifs: D(F/Y)DGD corresponding to the active site in RNA polymerases, DGD(G/A)D in some kinds of mutases, and DAKVGDGD in dihydroxyacetone kinase, where all three classes manipulate phosphate groups required for nucleotide metabolism in the very first stages of life. The results also support the idea that the [GADV]-amino acids, which accumulated at meaningfully high amounts on the primitive Earth, were used for synthesis of the most primitive proteins, leading to the emergence of life.

### 4.2. The Most Ancient Event Deduced Using the GADV Hypothesis (Top-Down Approach)

As described in Section 3, the GADV hypothesis is an idea mainly based on the GNC primeval genetic code hypothesis, which was obtained by analysis of databases of genes and proteins, starting from studies on the formation mechanism of entirely new genes in presently existing microorganisms. So, the most ancient event deduced from the GADV hypothesis is the establishment of GNC primeval genetic code encoding four [GADV]-amino acids (Figure 5). This means that the GNC code encoding [GADV]-amino acids can be used as the common event deduced from the bottom-up approach, when the consistent evolutionary process from the birth of the Earth through to the emergence of life and the modern life system is considered.

### 4.3. Deduced Evolutionary Steps from the Formation of Earth to the Emergence of Life

As described above, the accumulation of [GADV]-amino acids and the establishment of the first genetic code or GNC primeval genetic code encoding the amino acids are assumed as the common event supposed from the results, which were obtained from two counter-directional (bottom-up and top-down) approaches. Thus, the consecutive evolutionary steps from the accumulation of simple organic compounds on the primitive Earth to the emergence of life can be rationally deduced by using a common event as a juncture, as shown in Figure 7.

**Figure 7 life-06-00006-f007:**
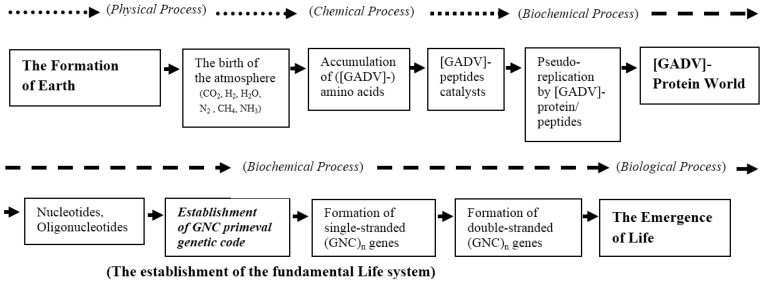
Evolutionary steps from the formation of Earth to the emergence of life, deduced from the bottom-up approach and GADV hypothesis (the top-down approach). Periods of physical-chemical, biochemical, and biological processes are shown by dotted, broken, and solid arrows, respectively. A juncture (the establishment of GNC primeval genetic code), which is used to determine the evolutionary steps that are consistent between the bottom-up and the top-down approaches, is indicated by bold italic letters.

## 5. Discussion

As repeatedly mentioned in this review article, the most important step in solving the riddle of the origin of life is to make clear the process by which the fundamental life system, composed of genes, genetic code, and proteins, was established on the primitive Earth. Now, I think that the evolutionary steps from simple inorganic and organic compounds as CO_2_, H_2_O, H_2_, N_2_, and CH_4_ to the modern life system have come into considerably clear view (Figure 7). However, there are still some other difficult problems that must be overcome to explain the evolutionary process to the emergence of life.

### 5.1. Why Can [GADV]-Amino Acids Accumulate by Prebiotic Means?

If a product, such as one of the [GADV]-amino acids, which was absent at the time, was synthesized with simpler chemical compounds, which were present in high amounts at the time, the product should gradually accumulate according to the law of chemical equilibrium, although it is necessary for reactants to receive sufficient energy to pass through an energy barrier of the physical or chemical reaction and for the product to have sufficiently high stability under the condition. In addition, random processes with prebiotic means generally cause both synthesis and degradation of organic compounds. Even in such a case, [GADV]-amino acids leading to the emergence of life could be accumulated on the primitive Earth. The reason is because accumulation of the amino acids could be observed as a result of a one-directional phenomenon, as [GADV]-amino acids are sufficiently stable. The accumulation could be further accelerated if the amino acids were excluded from the reaction system, for example by drying the products. Thus, a repeated heat-drying or dehydration process in depressions of rocks on seashores and tide pools on the primitive Earth could lead to accumulation rather than disappearance of the amino acids. So, the amino acids could easily accumulate under the circumstances on the primitive Earth in order of simple chemical structure of amino acids, as Gly [G], Ala [A], Asp [D], and Val [V].

The stronger the synthetic activity of a chemical compound becomes, the faster the product accumulates, even when synthesis and degradation of the product proceed with prebiotic means in parallel. Therefore, formation of a system more efficiently producing a product could lead to the development of an evolutionary step toward the emergence of life, in the order of simpler and more imperfect systems to more complex and more perfect systems, step by step.

### 5.2. Why Can Water-Soluble Globular Proteins Be Formed in the Absence of Any Genetic System?

I have defined protein 0^th^-order structure as a specific amino acid composition in which water-soluble globular proteins, a prerequisite for high catalytic activity, can be produced at a high probability even by random polymerization of amino acids [51,57,58,61]. One of the protein 0^th^-order structures is a composition containing [GADV]-amino acids at a roughly equal ratio or about one-fourth each. Owing to the protein 0^th^-order structure, water-soluble globular proteins with weak but sufficiently high catalytic activity could be produced at a high probability even in the absence of genetic function or before creation of the first (GNC)_n_ gene. Evolutionary processes leading to the emergence of life through the formation of the [GADV]-protein world always proceeded to produce more perfect [GADV]-proteins than those one step before, under the protein 0^th^-order structure.

For example, the appearance of [GADV]-peptides/proteins as aggregates could promote biochemical reactions to accumulate various organic compounds including [GADV]-peptides and [GADV]-proteins themselves. As a result, the [GADV]-protein world was formed by pseudo-replication, with [GADV]-peptides/proteins having catalytic activity for peptide bond formation. The most primitive metabolic system was formed in the protein world. [GADV]-peptides/proteins with high substrate selectivities for nucleobases, ribose, and phosphate could give rise to syntheses of nucleotides and oligonucleotides.

Accumulation of oligonucleotides containing GNC led to the establishment of the first genetic code, GNC. The establishment of the GNC code made it possible to synthesize more perfect [GADV]-peptides and/or [GADV]-proteins containing roughly equal amounts of [GADV]-amino acids and to exclude non-natural amino acids and compounds other than [GADV]-amino acids.

Furthermore, the first single-stranded (GNC)_n_ gene was formed by joining two GNC triplets in complexes of oligonucleotide-containing GNC with one [GADV]-amino acid. This made possible the synthesis of a number of [GADV]-peptides/proteins with the same amino acid sequence. It is important to understand that [GADV]-proteins or aggregates of [GADV]-peptides, not merely [GADV]-peptides, would be required for synthesis of (GNC)_n_ gene, because two nucleotides must be bound to the proteins to form a phosphodiester bond between two nucleotides.

Finally, ds-(GNC)_n_ genes were produced by [GADV]-proteins with high catalytic activities in the protein world. The formation of the first ds-(GNC)_n_ gene made it possible to produce homologous genes and to create entirely new genes from sense and antisense sequences of the gene after gene duplication, respectively. With the formation of ds-(GNC)_n_ gene, the replication system was established and the first life could emerge through acquisition of various kinds of genes and of the replication system. Therefore, it is rationally assumed that the fundamental life system was formed from the primitive system as going up a spiral staircase, and water-soluble globular proteins were always produced under the protein 0^th^-order structure. Transcription and translation of ds-(GNC)_n_ genes could presumably be carried out by a [GADV]-protein (a primitive RNA polymerase) and an oligonucleotide-[GADV]-protein complex (a primitive ribosome).

### 5.3. Life Could Emerge with Some Good Luck and Trial and Error

Of course, random processes did not always lead directly to the formation of the life system. Various kinds of chemical reactions would randomly occur in an innumerable number of depressions of rocks and tide pools to produce various organic compounds independently of the emergence of life. GNC primeval genetic code could be successfully established and the first double-stranded (GNC)_n_ genes could be fortunately produced in only one depression of a rock or a tide pool on the vast primitive Earth. As a result, the first life could emerge by chance in a depression or tide pool, as one of countless reaction systems. Chemical compounds in other depressions and tide pools could be used as nutrients for the first life and its descendants.

I have considered the steps to the emergence of life as above, but it is important to confirm experimentally whether the idea is correct or not. Thus, experiments are undoubtedly essential to elucidate how life emerged on the planet. However, it is also true that it would be impossible to solve the riddle of the origin of life only from experiments based on the bottom-up approach (Figure 1). So, it must be essential to solve the riddle using two counter-directional (bottom-up and top-down) approaches, which are based mainly on experiments and computational analyses of databases of genes and proteins or theoretical considerations, respectively.

## 6. Conclusions

Until now, researchers have tried to understand, using the bottom-up approach, how the first life emerged on the primitive Earth, but it has been so far proven impossible to solve the riddle of the origin of life. It is principally impossible to elucidate the formation process of the fundamental life system composed of genes, genetic code, and proteins. On the other hand, I started with a study on the formation of entirely new genes in presently existing microorganisms, and finally reached the conclusion that life emerged not from the RNA world but from the [GADV]-protein world ([GADV]-protein world hypothesis or GADV hypothesis). Therefore, I accidentally adopted a top-down approach for solving the riddle. Furthermore, I noticed that the riddle of the origin of life could be solved by using a common event, which was obtained by the two counter-directional approaches as a juncture.

The probable evolutionary steps from the formation of the Earth to the emergence of life, which were deduced from the results obtained with the two approaches, are as follows: (1) The primitive atmosphere composed of CO_2_, H_2_, H_2_O, N_2_, CH_4_, NH_3_, *etc.* was formed. (2) Simple amino acids, such as glycine [G], alanine [A], aspartic acid [D], and valine [V], were physically and chemically synthesized and accumulated on the primitive Earth. (3) Peptide catalysts, such as Gly-Gly, Gly-Asp, and [GADV]-peptides, were produced. (4) The [GADV]-protein world was formed by pseudo-replication with [GADV]-peptides/proteins. (5) Nucleotides and oligonucleotides (RNA) were synthesized and accumulated in the protein world. (6) GNC primeval genetic code encoding [GADV]-amino acids was established. (7) Single-stranded (GNC)_n_ gene(s) and successively double-stranded (GNC)_n_ gene(s) were formed. (8) Finally, the first life emerged on the primitive Earth.

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
