# Peer review of "Evolutionary Steps in the Emergence of Life Deduced from the Bottom-Up Approach and GADV Hypothesis (Top-Down Approach)"

_life, 2016, doi:10.3390/life6010006_

Round 1
Reviewer 1 Report
The presented review by Ikehara is a strong supporter of the GADV peptide world hypothesis - one of several hypothesis on the emergency of life. The well written review thereby points out that both approached from buttom up (such as synthesis of small replicative molecules) and from top down (such as analyzing the use of codon in organisms from today) hint that the first replicative elements, that yielded life, are peptides composed of the amino acid residues G, A, D, or/and V. It is thereby stressed that whether one starts with the RNA hyopthesis or the peptide hypothesis, the two systems of interest may had to come close in space.
As a minor point, it is suggest to add some references on amlyoids and the amyloid world hypothesis. Amyloids are peptide aggregates, which can be enzymatic active, GADV peptides are able to form amyloids, the genetic code appears to evolve away from amyloid sequences, amyloids can replicate themself under some conditions, etc. It has been suggested that the amyloid fold is the first fold.
Author Response
The author is very thankful to the reviewer 2 for giving some constructive comments and suggestions to improve the manuscript.
As a minor point, it is suggest to add some references on amlyoids and the amyloid world hypothesis.
Response: I add two references on amyloids and the amyloid world hypothesis in Introduction of the revised manuscript, as suggested by referee 1.
Thank you very much.
Reviewer 2 Report
The manuscript reviews research fundamentally related to the GADV hypothesis proposed by the author. The manuscript is poorly written and very difficult to follow. Many concepts are introduced without adequate description. For example, protein formabilities, sense and antisense of GC-NSF(a), and other concepts/experiments introduced in section 5.1 onwards that are necessary to understand the GADV hypothesis are not explained in understandable terms. This makes further explanations of biases of the genetic code, peptides, and supporting evidence of the main hypothesis almost impossible to follow. I suggest a major revamping of the entire manuscript. I also suggest a more balanced description of background knowledge related to this 'bottom up' and 'top down' link that the author wants to stress. Even the pros and cons of the RNA world are poorly explicated, especially in light of very recent harsh critique and experimental evidence (e.g. Kurland, Carter, Caetano-Anoles, Wächstershäuser). No description of studies that have explored biases in amino acid composition of proteins and their structures are reported. Also, the use of the word 'evolutionary' to describe chemical and biochemical experiments in page 4 for example is misleading. The work that is presented in this manuscript relies almost exclusively on prebiotic chemistry and predictive science. There are no attempts to reconstruct history in this paper. In other words, the 'top-down' approach is not phylogenetic, but predictive based on attempts to recreate conditions extisting on early Earth (truly bottom-up) matching an hypothesis supported by statistical patterns in a small sliver of extant molecules. How reliable are these patterns? What efforts have been invested to make analyses more comprehensive? What about molecular resurrection? What about biochemistry, enxymology and atomic structure? In conclusion, the review article deserves considerable additional effort to make underlying ideas effective and understandable.
Author Response
The author is very thankful to the reviewer 3 for giving some constructive comments and suggestions to improve the manuscript. The following responses are some revision explanations.
Response:
1. As indicated by the referee 3, many sentences were added in the manuscript to explain the concepts as protein formabilities, sense and antisense of GC-NSF(a) and concepts/experiments, in Section 3.1 of the revised manuscript.
2. In addition, I looked at the original manuscript again and revamped the whole manuscript as much as I can.
3. I added some papers, which Kurland, Carter, Caetano-Anoles, and Wachtershauser described harsh critique against RNA world theory, in the references of my manuscript.
4. The word ‘evolutionary’ was exchanged with the word ‘evolution’ to avoid misleading.
5. I consent with comments of reviewer 3, saying that the 'top-down' approach is not phylogenetic, but predictive. But, I have described in Section 3.5 of the revised manuscript (in Section 5.7 of the original manuscript) honestly that only a few experimental results have been obtained thus far. But, I would like to stress, describing in the Same Section, that the hypothesis is not a purely theoretical idea, since the hypothesis was based on databases, which were obtained by experiments, and is based on protein 0th-order structure as [GADV]-amino acids, which satisfies the four structural conditions obtained from analysis of experimental data of proteins in extant micro-organisms.
Reviewer 3 Report
See attachement.

Author Response
The author is very thankful to the reviewer 1 for giving some constructive comments and suggestions to improve the manuscript.
1. The slant of the manuscript, particularly in the first part, is in the form of a review, but for the readers of this LIFE issue, devoted to specialists, is not necessary to repeat the concepts and points that every student knows. This is also the case for the section on the RNA world. No need to repeat what everybody knows.
Response: As suggested by referee 2, I deleted unnecessary descriptions such as Fig. 1 and Section 3 in the original manuscript for specialists working in the field of life science.
2. This is particularly true for the abstract. this is not appropriate, as one should concisely state there what the paper is about, avoiding general, well known considerations . Just say in the abstract what the paper is about and what is the main message.
Response: I rewrote the abstract extensively as suggested by referee 2.
3. The paper needs a close editing by an English mother language editor, the present form is not really sufficient.
Response: I asked to someone, who is an English mother language editor, to revise my English, please check.
4. The manuscript in the present form is also too long, and repetitious. For example, the conclusion contains statements and concepts which have been said before in the text, almost with the same language. This long part can be reduced to a concluding remarks section, where one summarises the key points in a very concise manner.
Response: I omitted repetitious parts such as Section 5.5 and rewrote extensively the whole manuscript as suggested by referee 2. According to the rewriting, several Figures 1, 5, 7 and 11 were omitted from the original manuscript, too.
5. All those expressed above are formal –but very important-points. There is also a basic point
about the fundament of the all concept. This is about the primitive genes, which in principle
should code for the GADV proteins. The commonly accepted notion of codifying genes is based on a transcription-translation mechanism, which is operated by several specific enzymes; and there is the notion of ribosomes, as the place where translation takes place.
And in the scheme proposed by the author, there is nothing about this point, and it is then not
clear how one can go from a primitive set of genes to the proteins. No transcription, no
messenger RNA, and no enzymes for the translation? Or the Author assumes that is already all there? If the author does not need all that in his theory, it would be great-but then this should be well clarified.
Response:
I have considered that even the primitive gene, as (GNC)n gene, would be transcribed by a protein, or a [GADV]-protein, and translated with primitive tRNA and primitive ribosome. So, I added some sentences in the last part of Section 5.2 in the revised manuscript to explain my consideration in the manuscript.
Thank you vey much.
Round 2
Reviewer 3 Report
The manuscript of Ikehara is now acceptable.